# Accumulation mechanisms of radiocaesium within lichen thallus tissues determined by means of *in situ* microscale localisation observation

**Terumi Dohi**[1]*, **Kazuki Iijima**[1], **Masahiko Machida**[2], **Hiroya Suno**[2¤], **Yoshihito Ohmura**[3], **Kenso Fujiwara**[1], **Shigeru Kimura**[4], **Futoshi Kanno**[5]

**1** Sector of Fukushima Research and Development, Japan Atomic Energy Agency, Miharu-town, Fukushima, Japan, **2** Center for Computational Science & e-Systems (CCSE), Japan Atomic Energy Agency, Kashiwa-city, Chiba, Japan, **3** Department of Botany, National Museum of Nature and Science, Tsukuba-city, Ibaraki, Japan, **4** Nuclear Engineering Co., Ltd., Ibaraki, Japan, **5** Pesco Co., Ltd., Minato-ku, Tokyo, Japan

¤ Current address: Research Center for Computing and Multimedia Studies, Hosei University, Koganei-city, Tokyo, Japan
* dohi.terumi@jaea.go.jp

**Data Availability Statement:** All relevant data are within the paper and Supporting information.

## Abstract

Many lichens are well known to accumulate radiocaesium and, thus acting as biomonitors of contamination levels. However, the actual localisation and chemical forms of radiocaesium in contaminated lichens have not yet been elucidated because, despite their high radioactivity, these forms are present in trace amounts as chemical entities. Here, we use autoradiography and demonstrate for the first time *in situ* microscale localisation of radiocaesium within thallus tissues to investigate the radiocaesium forms and their accumulation mechanism. Radiocaesium distributions showed similar trends in lichen tissues collected two and six years after the Fukushima nuclear accident. The radiocaesium was localised in the brown pigmented parts i.e., melanin-like substances, in the lower cortex of lichen thallus. Quantum chemical calculations showed that functional group of melanin-like substances can chelate $Cs^+$ ion, which indicates that the $Cs^+$ ions form complexes with the substances. Based on these findings, we suggest that radiocaesium ions may be retained stably in melanin-like substances for long periods (two to six years) due to steric factors, such as those seen in porphyrin-like structures and *via* multimer formation in the lower cortex. In addition, electron microscopy and autoradiography were used to observe radiocaesium-bearing microparticles (CsMPs) on/in the upper cortex and around the medullary layer. Micron-sized particles appeared to adhere to the surface tissue of the thallus, as shown by electron microscopy, suggesting that the particles were trapped by development of an adhesive layer; that is, CsMPs were trapped both physically and physiologically. These findings provide information on *in situ* localisation of two chemical forms of radiocaesium, cations and particles, in lichen thallus tissues and their accumulation mechanisms.

**Funding:** This work was financially supported by an internal budget of Japan Atomic Energy Agency for the Fukushima reconstruction research activity. This research was also supported by Grant-in-Aid for Challenging Exploratory Research operated by TD (KAKENHI) (no.16K12627) from the Japan Society for the Promotion of Science. The funders had no role in study design, data collection and analysis, decision to publish, or preparation of the manuscript. Nuclear Engineering Co., Ltd. employed SK, and Pesco Co., Ltd. employed FK, these commercial affiliations did not play a role in this study.

**Competing interests:** The authors have declared that no competing interests exist. Nuclear Engineering Co., Ltd. and Pesco Co., Ltd. which are author's affiliations, do not alter our adherence to PLOS ONE policies on sharing data and materials.

## Introduction

Lichen is a symbiotic organism composed of fungus and alga/cyanobacterium. Lichens are widely distributed in terrestrial ecosystems from the Tropics to the Polar regions. It is well known that lichens accumulate high concentrations of heavy metals (e.g., Cu, Fe, U) [1–4] and radionuclides (e.g., $^{210}$Pb, $^{210}$Po, $^{238}$U, $^{137}$Cs) [1, 5, 6]. Moreover, they grow on stable substrates, such as tree bark, rock, soil, etc, for over decades, do not have a root system like vascular plants, and are nourished by photosynthesis carried out by symbiotic algae. Mineral nutrients are incorporated directly from the environment (e.g. *via* dry deposition and wet deposition) through the entire thallus [7–9]. Because of this, lichens have been examined as biomonitors of metal contamination levels, including those resulting from radionuclide contamination [10–12]. In particular, radiocaesium ($^{137}$Cs) has been a focus for biomonitoring since the nuclear-weapon tests of the 1960s, and has been applied to study other nuclear events, such as the Chernobyl nuclear accident (26 April 1986) and the Fukushima nuclear accident (11 March 2011), primarily because of its predominance among radioactive contaminants and its long half-life (30.1 *a*) [1, 6, 11, 13–15]. Sloof and Wolterbeek [16] showed that the $^{137}$Cs activity concentrations found in *Parmelia sulcata* six months after the Chernobyl accident exhibited a 1:1 ratio with the fallout amount per area. Even two years after the Fukushima accident, a correlation was shown between $^{137}$Cs activity concentration in parmelioid lichens and $^{137}$Cs deposition in soil (as a proxy for fallout amount) [17].

The biological half-lives of $^{137}$Cs in lichens vary from ~1 *a* to 17 *a*, with a wide range summarised by Anderson *et al.* [12]. This seems to be due to the different physicochemical states of Cs, which result in different balances for Cs entering and leaving the lichen after excluding other factors such as growing conditions, surface area and biomass [12]. However, the mechanism for Cs accumulation is not well understood [18].

The mechanisms for accumulation of metals in lichens have been categorised as follows: (i) extracellular absorption (ion exchange), (ii) intracellular absorption, and (iii) particle capture [8, 19]. Nedić *et al.* [20] performed *in situ* fractionation experiments in radiocaesium accumulation study and suggested that biomolecules might be relevant to radiocaesium retention, although their chemical characteristics (constitution and molecular weight) were not known. Dohi *et al.* [21] isolated particles indicated spot-like distributions on the autoradiographs from parmelioid lichens and determined that they were CsMPs. Nonetheless, it remains unclear where and in what form radiocaesium is present in lichens. Knowledge of this would lead to an understanding of different types of radiocaesium accumulation mechanism and enable estimates of fallout amounts depending on different deposition forms. Autoradiography is an easy and useful method for understanding the two-dimensional distribution, uniform distribution and spot-like distribution of radiocaesium, and may therefore be able to distinguish between particulate Cs and other forms of Cs [22].

The substances contributing to Cs retention may be determined from the locations of Cs. Lichens produce various compounds, e.g., metabolic substances, which are suggested to form metal complexes *via* chelation [23, 24]. Brown to black pigmented substances, melanin-like substances, within lichen tissues have also been found to contain heavy metals (e.g., U, Cu, Fe), but it is unclear whether the heavy metals are chemically bound [2, 3, 25]. In studies of fungi, the melanin pigment was considered essential for absorption and retention of radionuclides (e.g., $^{137}$Cs and $^{60}$Co) since high melanin precipitation was found in spores exhibiting high radionuclide uptake [26, 27]. Suno *et al.* [28] successfully evaluated stable forms of Cs$^+$ ions complexed with metabolic substances common in parmelioid lichens such as *Flavoparmelia caperata* (L.) Hale and *Parmotrema tinctorum* (Nyl.) Hale: oxalic acid, atranorin, lecanoric acid, usnic acid and protocetraric acid by applying quantum chemical

calculations. These lichen metabolites are secreted at specific locations in the thallus tissue, such as atranorin in the upper cortex, lecanoric acid in the medulla, and brown (or black) melanin-like substances in the lower cortex [29, 30]. Therefore, understanding the locations of $^{137}$Cs accumulation in thallus tissue is expected to reveal the substances involved in the mechanism of Cs accumulation.

Thus, this study was designed to investigate the differences in radiocaesium forms in lichen thallus tissues based on the distribution of radioactivity observed by autoradiography. Then, we analysed thin tissue sections of the tissue layers containing Cs by using autoradiography measurements and microscopic analyses, and determined for the first time the *in situ* micro-scale localisations of radiocaesium and the substances contributing to Cs accumulation. Based on these results, we discuss herein the mechanisms of radiocaesium retention in lichens.

## Materials and methods

### Samples

We selected two lichen samples from our previous studies of *Parmotrema tinctorum* (Parmeliaceae, Ascomycota) which were collected two and six years after the Fukushima nuclear accident on 11 March 2011; these were named "FY2012PT" and "FY2017PT", respectively, and exhibited similar $^{137}$Cs activity concentrations (1 130 Bq g$^{-1}$) [17, 22]. The samples were collected from the tree trunks of *Cerasus* sp. found growing within Okuma town. All sample identification data, e.g., detailed location and sampling date, are described in Dohi *et al*. [17, 22]. An individual of *P. tinctorum* was used as a background sample (named "B.G. PT" in this study) and was collected on 29 September 2016 on the premises of the Japan Atomic Energy Agency in Toki city, Gifu prefecture, which is located in the middle of Japan (ca. 410 km southwest of the Fukushima Dai-ichi Nuclear Power Station (FDNPS): 35°23′38.89″ N– 137° 11′24.89″ E). In addition, one CsMP sample isolated from the parmelioid lichens in Dohi *et al*. [21] and its radioactivity was used as a reference for autoradiography and for estimating $^{137}$Cs radioactivity (the radioactivity was already determined to be 3.95±0.02 Bq for $^{137}$Cs, corrected to 11 March 2011). This CsMP is referred to as "FY2016CsMP" in this study based on the sampling fiscal year.

### Autoradiography of lichen samples and selection of thallus tissue

As an initial screening analysis, autoradiography was used to visualise the radiocaesium distribution in the lichen samples and determine which tissue sections should be prepared for further analysis. First, the samples were attached to a BAS-IP imaging plate (SR 2040, Fuji Film, Tokyo, Japan). The exposure times ranged from 2 to 16 hours according to the samples and analytical purposes. For instance, 16 hours was applied to visualise the overall radiocaesium distribution in the lichen samples, and two or three hours to check for strong spot distributions during the fragmentation process before preparation of tissue sections. The autoradiography images of the imaging plate (IP) were acquired using a laser scanner (Typhoon FLA7000, GE Healthcare, Little Chalfont, UK) at a maximum resolution of 25 μm per pixel. Photostimulated luminescence (PSL) signals were also measured using the scanner. The signal spectral shapes were determined using imaging software (Digital Micrograph GMS3, free software, https://www.gatan.com/products/tem-analysis/gatan-microscopy-suite-software, Gatan, California, US). The size required for the fragmentation of lichen samples was approximately 2 × 2 mm area. By checking the signal intensity and spectral shape in the fragmentation area, a type of distribution was confirmed. Those with a constant range of variation in the signal intensity were defined here as having a "broad distribution". In contrast, the spectral shape with which a peak detected was defined as the "spot distribution". Finally, the thallus tissues

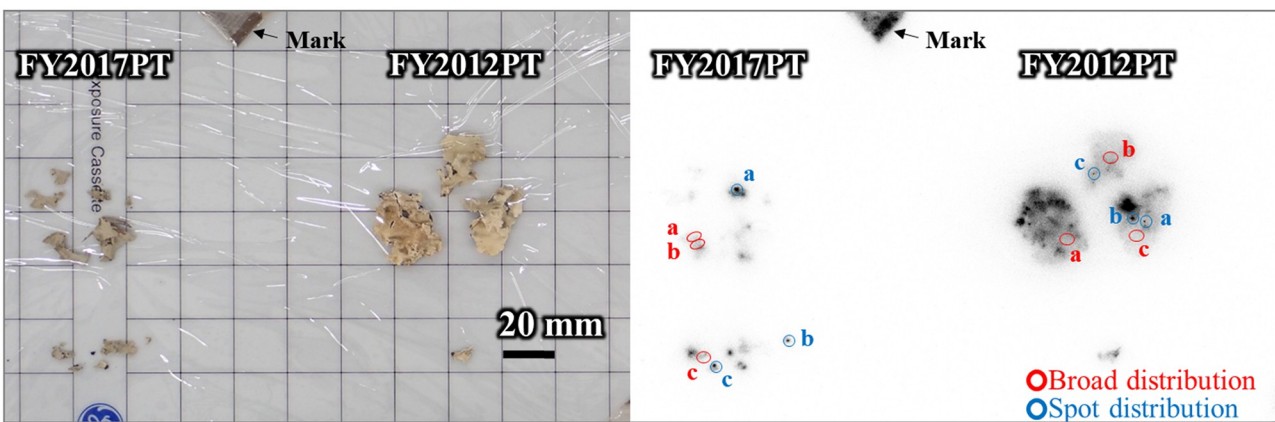

**Fig 1. Radiocaesium distributions in *P. tinctorum* collected in FY2012 and FY2017.** Autoradiograms. Left panel: images of FY2012PT and FY2017PT lichen samples. Right panel: autoradiograms of samples after 16 hours of exposure. Red circles and blue circles indicate typical broad distribution and spot distribution patterns, respectively. Each area marked "a" was used for preparation of the tissue sections.

selected for further analyses were determined based on the following: radiation distribution forms, PSL signal intensities, and signal spectral shapes (Figs 1 and 2).

**Preparation of thallus tissue sections.** The selected thallus tissue was prepared from sample fixation to resin embedding with a preparation protocol customised according to the preparative techniques used for ultrastructural studies [31]. The lichen samples were cut by a razor blade into small pieces of ca. 1 mm in width, fixed with 2% glutaraldehyde in 100 mM sodium phosphate buffer (pH 8.0), and kept at 4°C in a refrigerator for 2 hours. The fragments were immersed three times in sodium phosphate buffer solution at room temperature for 10 min each time, dehydrated in 100% ethanol three times for 20 min each time and immersed in propylene oxide (PO) three times for 10 min each time at room temperature. A resin was

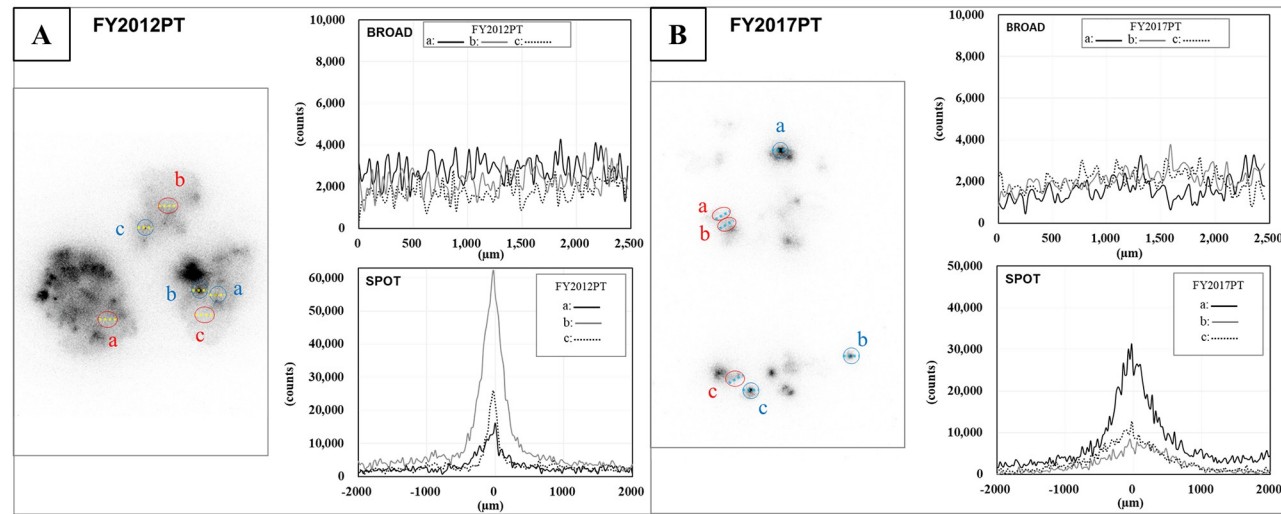

**Fig 2. Signal intensity patterns of the typical distribution examples in the areas of the FY2012PT and FY2017PT sample shown in Fig 1.** (A) FY2012PT sample. (B) FY2017PT sample. Left panel: line scan position (yellow dotted line in A, aqua dotted line in B) in the images for autoradiogram acquisition. Right panels: linear intensities of the selected areas in the left panel. Upper right panel: "broad distribution"; lower right panel: "spot distribution". The intensities ranged from ca. 400 to 4 000 in the broad distribution. The x-axis shows the linear distance of the distribution (μm), and the y-axis shows the PSL signal intensity value (counts) in both graphs. The x-axis is aligned with 0 in the spot distribution.

prepared with the following proportions: 48:19:33:2 for EPOK resin:DDSA hardener agent: MNA hardener agent:DMP-30 hardener accelerator. To replace the PO within the specimens with the prepared resin, the fragments were sequentially immersed in solutions with the different ratios, as follows: (2:1) PO:resin for 30 min, (1:1) PO:resin for 1 hour, and (1:2) PO:resin for 1 hour. They were then placed in resin overnight; all procedures were carried out at 4°C. The fragments in the resin were cut with a razor blade into pieces measuring ca. 0.5 × 0.5 mm and set in commercial flat embedding moulds. These moulds were placed in the oven and polymerisation was run at 60°C for 24 hours. The resin-embedded specimens were sliced into 5 μm-thick sections from the upper cortex to the lower cortex on an ultramicrotome (Leica EM UC7, Leica Microsystems, Tokyo, Japan) using glass knives (Leica EM KMR3, Leica Microsystems, Tokyo, Japan). The sliced sections were laid on a square-shaped cell plate in 3 rows ("a" to "c") and 25 columns ("1" to "25") in the order they were cut so that the location of each section in the thallus tissue was clear: This was identified as a combination of "row and column" layers (e.g., a-1 layer) in this study. Images of the sections were taken with a digital microscope at 200x magnification (VHX-8000, Keyence, Osaka, Japan). Each section was discriminated by this microscopic observation as the upper cortex (including algae), medullary layer and lower cortex (S1 Fig).

## Evaluation of the radiocaesium distributions in thallus tissue sections

The 5 μm-thick tissue sections were individually laid out and covered with plastic wrap for autoradiography. The tissue specimens with broad distributions were placed in a lead block enclosure during exposure to avoid outside radiation, and then they were exposed for 14 days. In the autoradiography, background intensity per pixel was calculated as an average intensity of pixels which were located away from lichen samples in the same autoradiogram. This background intensity was subtracted from that of each pixel attached to the lichen samples to obtain net intensity. For specimens with a spot distribution, the exposure times were 24 hours due to the high PSL signal intensities. The $^{137}$Cs radioactivities in the spot distributions within tissue sections were estimated using the PSL signal intensity of the "FY2016CsMP" sample as a reference [32].

## Quantum chemical calculations

To assess the stability of complex formation between Cs$^{+}$ ions and biological substances contained in tissues, where radiocaesium distributions were observed, quantum chemical calculations were performed. Since radiocaesium was localised in brown pigmented parts of the tissues resulting from melanin-like substances, as described in the result section, we selected the hydroxyindole monomer, i.e., indole-5,6-quinone, as the most representative structure in common melanin molecules that can form complexes with alkali-metal cations. The computational calculations were carried out by using exactly the same methodology as Suno *et al.* [28]. In brief, a neutral molecular model was created from the chemical structure of the monomer, and the free energy was calculated for formation of a stable complex with the Cs$^{+}$ ion in the aqueous phase. Our quantum computational technique used a stepwise approach. The first step combined the semiempirical PM6 (parametric model 6) method and the multicomponent artificial force-induced reaction (MC-AFIR) method to screen for stable reaction-product structures from among several candidates. The second step subsequently used density functional theory (DFT) to optimise the screened candidate structures at the *ab initio* quantum mechanical level. The third and final step combined DFT with the polarisable continuum model (PCM) to take into account solvation effects. In the stepwise calculations [33], MOPAC 2016 [34], GRRM14 [35], and Gaussian 16 [36] were used for high-speed semi-empirical

calculations, complex form explorations, and DFT structural-optimisation calculations with the PCM model, respectively.

## Electron microscopy of thallus tissue

Electron microscopy was performed to analyse elemental compositions of lichen tissue and particles on the tissue surface. Sample FY2012PT was cut into fragments measuring ca. 5 mm square by using a razor blade. The dorsal side of the thallus (upper cortex) was attached onto the vertical face of an aluminium sample table by carbon tape to assess the cross-sectional area (12.5 mm φ × 10 mm height). The ventral side of the other thallus fragment was attached onto the horizontal face of the table to observe the particles on the thallus surface. No conductive coating was deposited on these samples, as low-vacuum scanning electron microscope (SEM) was applied. Fragment observation and elemental analyses were carried out using a Schottky emission SEM (SE-SEM) (SU5000, Hitachi High-Tech, Tokyo, Japan) with an EDS system (Oxford, UK). The accelerating voltages applied were 15 kV for cross-section evaluation and 5 kV for particle observations on the thallus surface. Elemental distribution images of the cross-section were obtained with the EDS system.

**Analysis of caesium-containing particle.** A survey with electron microscopy mapping analysis was performed to search for Cs-containing particle on the lichen thallus surface. After cutting sample FY2012PT into square fragment measuring less than 5 mm square with a razor blade, the fragment was mounted onto a brass sample table (10 mm φ × 10 mm height) using carbon tape. To ensure conductivity of the sample, a ca. 16 nm-thick gold layer was coated onto the sample using gold ion sputtering (JFC-1500, JEOL,Tokyo, Japan). The analysis was performed as a 2-step survey using a field emission electron probe micro analyser (FE-EPMA) (JXA-8530F, JEOL) with an accelerating voltage of 15 kV under the following conditions: (1) at 500× magnification, a 320 μm square area was surveyed at 10 ms per point with 320 × 320 steps; and (2) at 5 000× magnification, a focus area where Cs was detected was surveyed at 1 ms (20 accumulations) per point with 320 × 240 steps. The target element, Cs, was discriminated by using wavelength dispersive X-ray spectroscopy (WDS) with FE-EPMA. Elements coexisting with Cs in the analysis areas were also examined.

## Gamma-ray measurements for radiocaesium

The radioactivities of $^{134}$Cs and $^{137}$Cs in the samples were measured by using a high-purity Ge-detector with a multichannel analyser. The methodology was the same as that described in Dohi *et al.* [21, 22]. Multiple fragments (6 pieces, 0.12 g) of the B.G. PT samples were obtained randomly by using a 10 mm diameter hole-punch and laid on a flat plate with a 42 mm diameter [22]. The measurement time was 24 hours, and the detection limits were $1.56 \times 10^3$ Bq kg$^{-1}$ for $^{134}$Cs at 605 keV and $2.62 \times 10^2$ for $^{137}$Cs at 662 keV. For the particle sample, a Cs-containing particle determined by FE-EPAM analysis was isolated from the thallus surface with a micromanipulator (AxisPro, Micro Support, Shizuoka, Japan). The radiocaesium ($^{134, 137}$Cs) activities were determined based on standard point sources, CZ402 for $^{134}$Cs and CS402 for $^{137}$Cs (Japan radioisotope association, Tokyo, Japan) [21]. The particle was measured for 302 400 s, and the radioactivities of $^{134}$Cs and $^{137}$Cs were decay-corrected to 11 March 2011.

## Results

### Radiocaesium distributions and their distinct patterns

Autoradiography was performed to visualise the radiocaesium distributions in the lichen thallus samples. Radiocaesium distributions are shown in Fig 1 for lichen thallus samples,

including partially homogenous (broad distribution), strong spot-like (spot distribution) and clustering forms. The broad distribution pattern was wavy and without prominent peaks, while sharp peaks were found for spot distributions (Fig 2). The PSL signal patterns and intensities for the broad and spot distributions in FY2012PT and FY2017PT are shown in Fig 2A and 2B, respectively. Their patterns and intensities were similar. The pattern for CsMP isolated from lichen is shown in S2 Fig. Radiation distributions were not detected for the B.G. PT sample. These results identified locations for preparing tissue sections with two typical distributions (broad and spot) under similar conditions with the samples FY2012PT and FY2017PT. In addition, this screening process was useful for comparing the radiocaesium locations within the lichen thallus, depending on the time elapsed since these two forms of radiocaesium were entrapped in the lichen.

## Radiocaesium localisation in thallus tissue sections

In the samples with broad distributions, significant radioactivity was detected in the c-3 to c-8 layers (7 sections: the c-8 layer contained 2 sections) within 51 sliced specimens of the FY2012PT sample (Fig 3). These sections all had brown pigmented tissue (Fig 4 and S3–S7 Figs). The radiocaesium localisations almost overlapped the brown pigmented parts of the lower cortex, although the localisations were slightly wider (Fig 4 and S3–S7 Figs). In FY2017PT, a similar localisation pattern was observed in which the radiocaesium localisations overlapped the pigmented parts from the b-4 to b-9 layers (6 sections) within 35 specimens

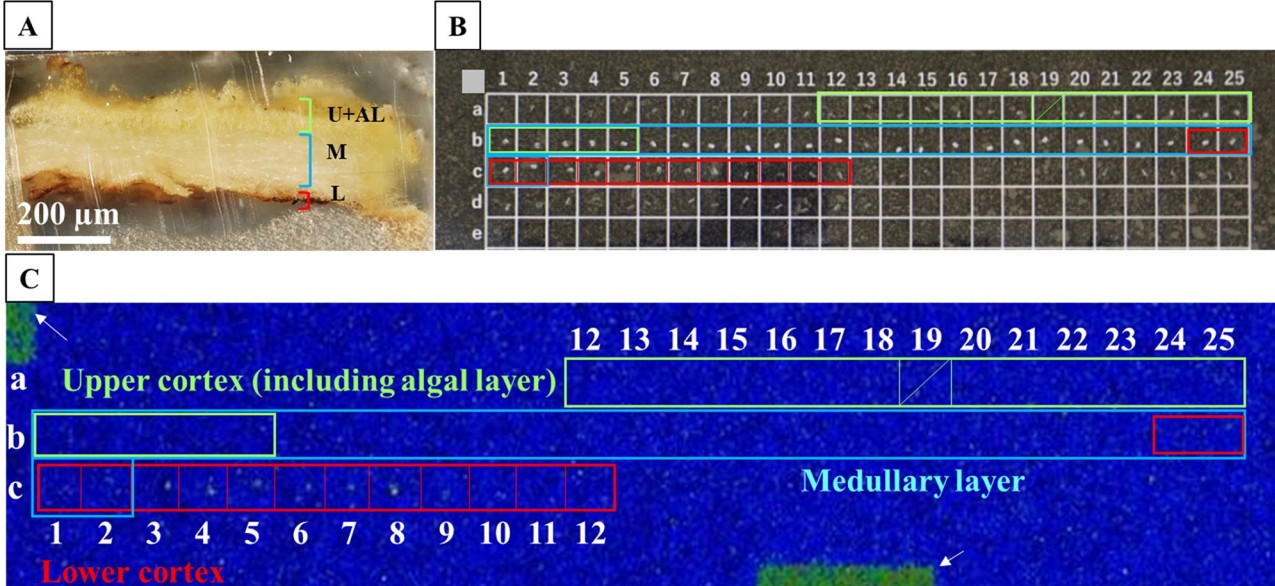

**Fig 3. Radiocaesium distributions in the tissue sections of the FY2012PT thallus selected from fragment in Fig 2 showing the broad distributions.** (A) Cross-sectional image of the FY2012PT lichen thallus tissue embedded in resin before microtome slicing. The thallus tissue was prepared from the broad distribution "a" of the FY2012PT sample in Figs 1 and 2. U+AL, upper cortex including the algal layer (these cannot be separated); M, medullary layer; L, lower cortex. (B) 5 μm-thick tissue sections of lichen thallus on a square-shaped cell plate. The tissue sections were sliced from the surface side of the thallus tissue towards the lower cortex layer and laid on the cell plate in that order. "a-1" represents the 1st layer from the surface side, and "b-1" represents the 26th layer. Coloured frames are sections containing tissue. Green, aqua and red boxes indicate the areas corresponding to the upper cortex (including the algal layer), medullary layer and lower cortex, respectively. Overlapping boxes indicate that two layers were contained in the frames. The boxed areas without a coloured frame, i.e., from "a-1" to "a-11", contained only resin and no tissue. A shaded area, i.e., "a-19", represents missing a number. The "c-8" layer contains 2 sections. (C) Autoradiograph images of each tissue section on the cell plate. Block numbers are the same as described in (B). White arrows indicate marks used to determine the locations of the cell plate boxes. The sections were exposed for 14 days on the imaging plate (IP). Significant radiocaesium distributions were detected in sections "c-3" to "c-8" corresponding to the lower cortex. However, two tissue sections were observed in "c-8" cell (See S7 Fig).

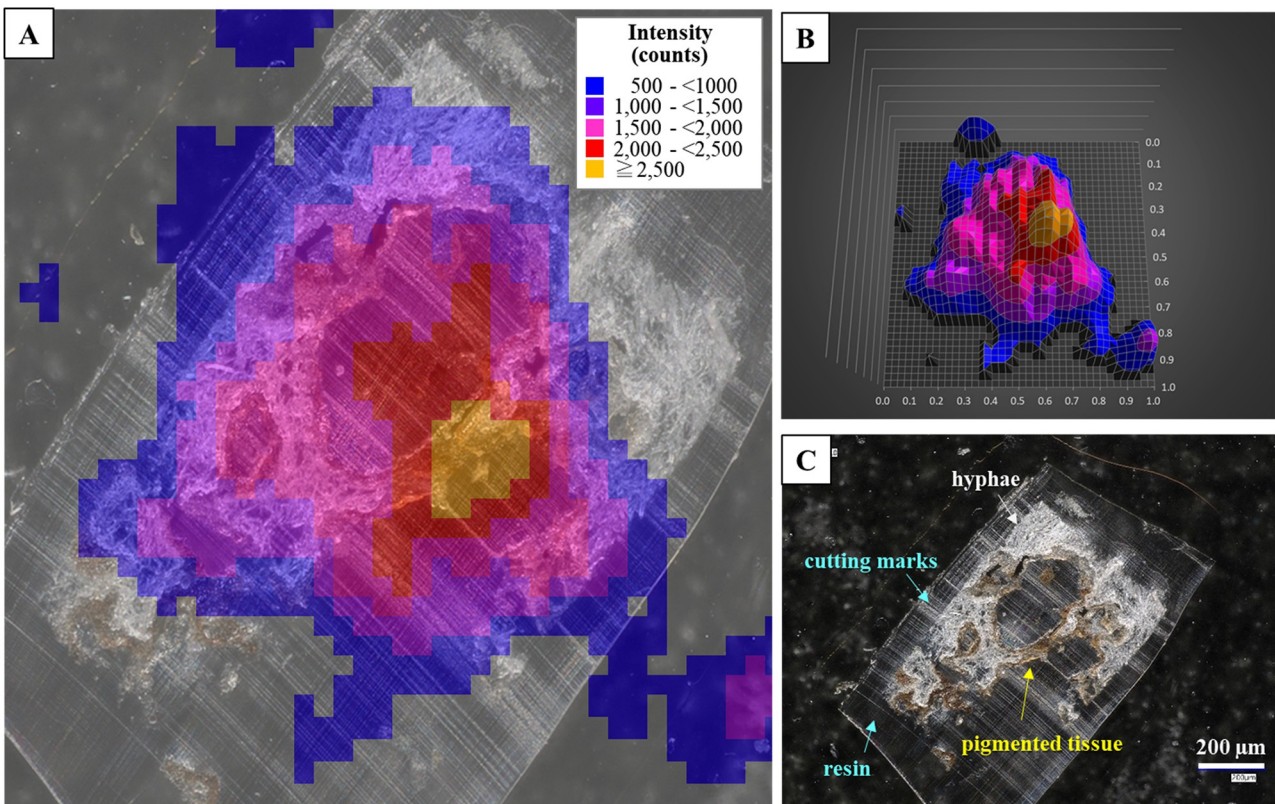

**Fig 4. Radiocaesium distributions in tissue section of the FY2012PT sample with "c-4" as an example.** (A) Overlapping image of the autoradiography result and tissue section image. The background counts were deducted from all count values. Each colour indicates the range of the PSL signal intensities, as shown in the legend, which also applies to (B). (B) Three-dimensional image of (A) in a 1 mm square area. (C) Image of tissue section taken with the digital microscope. The brown parts are pigmented tissue (yellow arrow), white parts are hyphae (white arrow), and transparent parts are resin (aqua arrow). Cutting marks from the glass knife appeared as a white linear pattern on the resin.

(the b-3 layer contained 2 sections) (Figs 5 and 6, S8–S12 Figs). Spot distributions were found in the a-21 and b-1 layers within the FY2012PT specimen and the c-1 layer within the FY2017PT specimen; these specimens were divided into 59 and 64 slices (Fig 7). The estimated radioactivities in those layers were 0.09 Bq (a-21), 0.29 Bq (b-1) and 1.93 Bq (c-1), respectively (Fig 8). The radiocaesium, which was distributed in a "broad distribution" was localised in the lower cortex in tissue sections from FY2012PT and FY2017PT. Spot distributions were found around the medullary layers in the FY2012PT and FY2017PT specimens. These results revealed the tissue layers in the lichen cross-section where the radiocaesium was localised. This location information was helpful for identifying the substances that interact with radiocaesium.

## Quantum chemical calculations

As radiocaesium was localised in the brown pigmented parts, i.e., melanin-like substances, of the lower cortex, quantum chemical calculations were performed to assess the stability of the complex formation between melanin-like substances and $Cs^+$ ions. Fig 9 shows the most stable structure for a complex formed between the indole group of the melanin monomer and a $Cs^+$ ion, as determined by quantum calculations under neutral conditions. The free energy for $Cs^+$ ion complexation in the aqueous phase, $\Delta G_{aq}^{cmplx}$, was calculated to be −4.31 kcal mol$^{-1}$. The distances from the $Cs^+$ ion to the O$^-$ atoms were 3.22 and 3.23 Å, respectively (Fig 9). This result

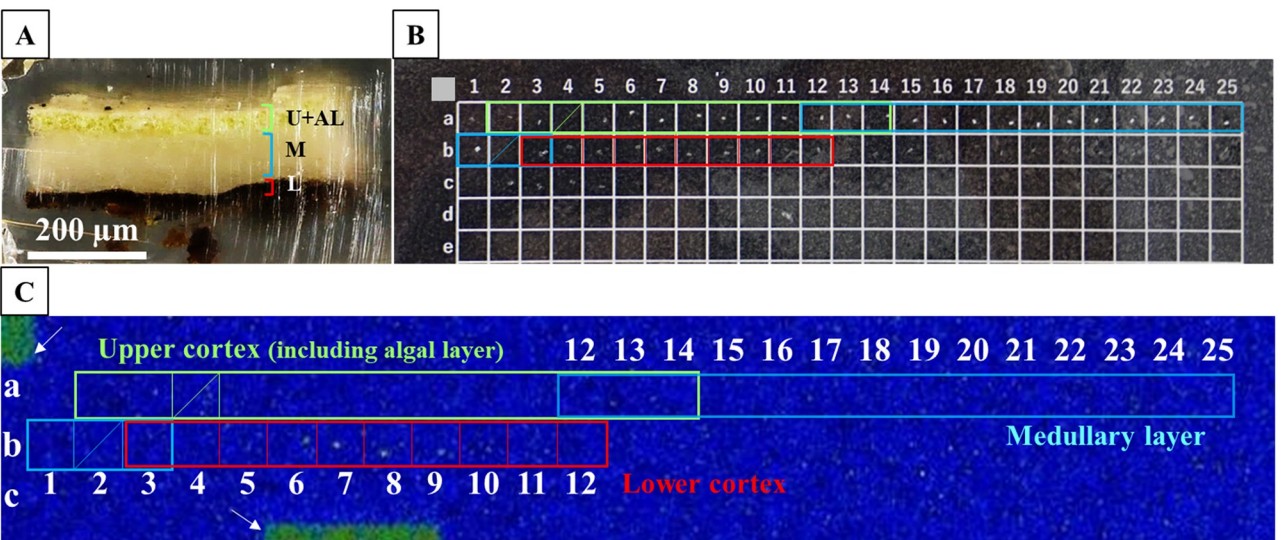

**Fig 5. Radiocaesium distributions in the tissue sections of the FY2017PT thallus selected from fragment in Fig 2 showing broad distributions.** (A) Cross-sectional image of the FY2017PT lichen thallus tissue embedded in resin before microtome slicing. The thallus tissue was prepared from the broad distribution "a" of the FY2017PT sample in Figs 1 and 2. U+AL, upper cortex including the algal layer; M, medullary layer; L, lower cortex. (B) 5 μm-thick tissue sections of lichen thallus on a square-shaped cell plate. Coloured frames are sections containing tissue. Green, aqua and red boxes indicate the areas corresponding to the upper cortex (including the algal layer), medullary layer and lower cortex, respectively. Overlapping boxes indicate that two layers were contained in the frames. The boxed areas without a coloured frame are empty sections containing only resin and no tissue. Shaded areas, i.e., "a-4" and "b-2", are missing numbers. The "b-3" layer contains 2 sections. (C) Autoradiograph images of each tissue section on the cell plate. Block numbers are the same as those in (B). White arrows indicate marks used to determine the locations of the cell plate boxes. The sections were exposed for 14 days on the IP. Significant radiocaesium distributions were detected in sections "b-4" to "b-9" corresponding to the lower cortex.

proves that $Cs^+$ ions can form complexes with a functional group of melanin-like substances in a neutral aqueous phase.

## Electron microscopic analyses of the cross-section and surface of thallus tissue

To investigate the relationships between localisations found for radiocaesium in tissue sections and those of other elements, the elemental distributions were examined by SE-SEM with EDS. Fig 10A shows a back-scattered electron (BSE) image of a tissue cross-section of the FY2012PT sample. Measuring approximately 30 μm thick from the surface in the upper cortex layer (including the algal layer), the 30 μm-tissue layers were conglutinated, and K was more uniformly accumulated in the cross-section than in other parts (Fig 10B). The middle part of the cross-section showed the medullary layer, which demonstrated an increase in brightness in the BSE image and was brighter than the upper and lower cortex layers due to a higher Ca distribution (Fig 10A and 10C). Ca (and/or its compounds) was distributed in the tissues and dominated elements with lower atomic numbers (i.e., C, H and O) (Fig 10A and 10C). The lower cortex layer showed more tissue density than the medullary layer (Fig 10A). Accumulations of other elements were not found in the lower cortex, where radiocaesium localisation was observed in the tissue sections.

The thallus surface was also examined by SE-SEM to determine the presence of micron-sized particles. Fig 11 shows a BSE image of the thallus surface of FY2012PT. Particles exhibiting sizes ranging from submicron to several tens of microns were found on the surface, and small pores and cracks were also observed. Micron-sized particles were partially buried in the surface tissue (Fig 11).

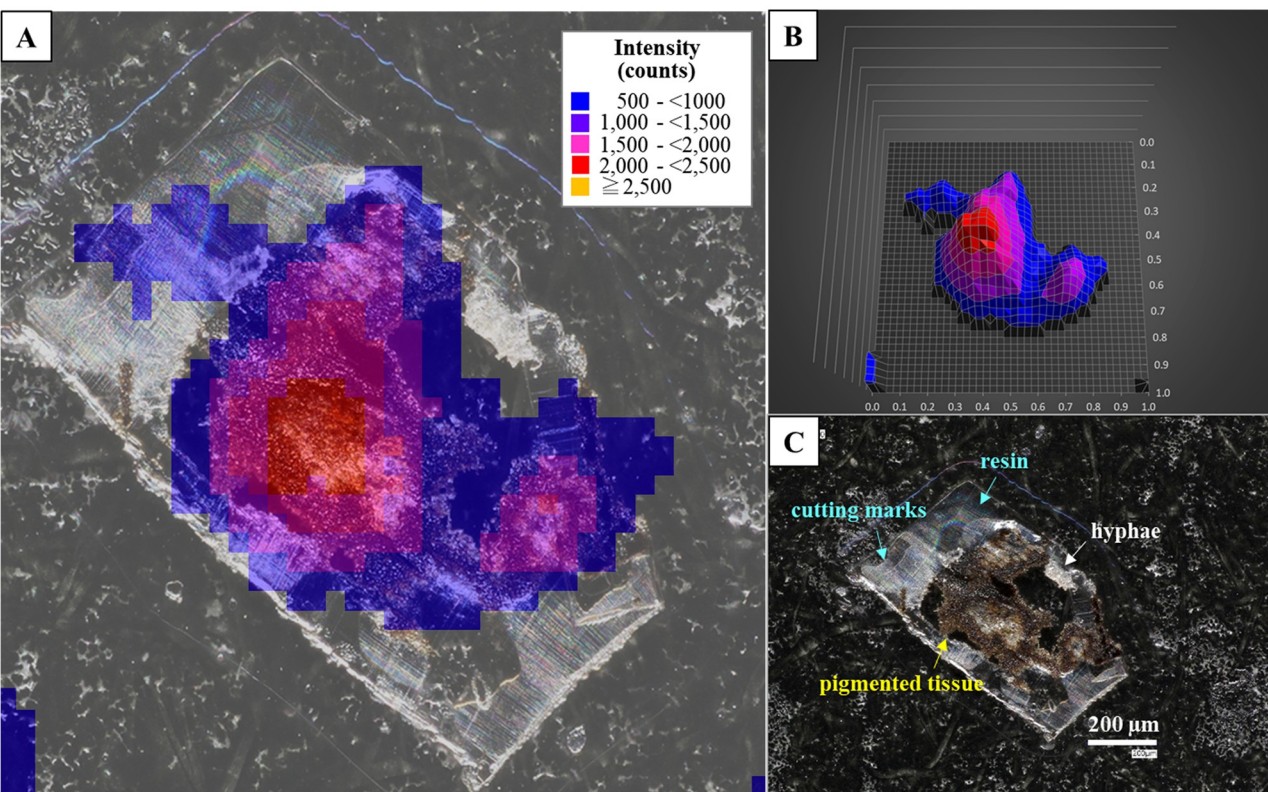

**Fig 6. Radiocaesium distribution in tissue section of the FY2017PT sample with "b-5" as an example.** (A) Overlapping image of the autoradiograph result and tissue section image. The background counts were deducted from all count values. Each colour indicates the range of the PSL signal intensities as shown in the legend, which also applies to (B). (B) Three-dimensional image of (A) in a 1 mm square area. (C) Image of tissue section taken with the digital microscope. The brown parts are pigmented tissue (yellow arrow), white parts are hyphae (white arrow), and transparent parts are resin (aqua arrow). Cutting marks from the glass knife appeared as a white linear pattern on the resin.

FE-EPMA analysis was carried out to determine Cs-containing particle in the vicinity of the spot distribution in the lichen sample. The spot distribution in the FY2012PT fragment was confirmed with a 3-hour autoradiography study. Cs-containing particle was identified on the thallus surface of the fragment by using a FE-EPMA analysis (Fig 12). The diameter of the Cs-containing particle was ca. 2 μm, as measured in the BSE image, and it also contained Si, O, Fe, Zn, Sn, Cl, etc. (S13 Fig). These results for size and for elements co-existing with Cs, were helpful in comparing the characteristics of CsMPs reported in previous studies.

## Radiocaesium activities determined by gamma-ray measurements

Radiocaesium activities were measured to examine the radiocaesium activity concentrations of the B.G. PT samples. In addition, it was measured to demonstrate whether Cs originated from the Fukushima accident in the isolated Cs-containing particle determined by FE-EPMA (Fig 12). Radiocaesium activities in the B.G. PT samples were below the detection limits for $^{134}$Cs and $^{137}$Cs. The activities for $^{134}$Cs and $^{137}$Cs in the isolated Cs-containing particle were 0.72 ±0.08 Bq and 0.71±0.01 Bq, respectively. In addition, the $^{134}$Cs/$^{137}$Cs activity ratio was determined to be 1.01±0.11. The Cs in the Cs-containing particle detected by FE-EPMA consisted of at least $^{134}$Cs and $^{137}$Cs.

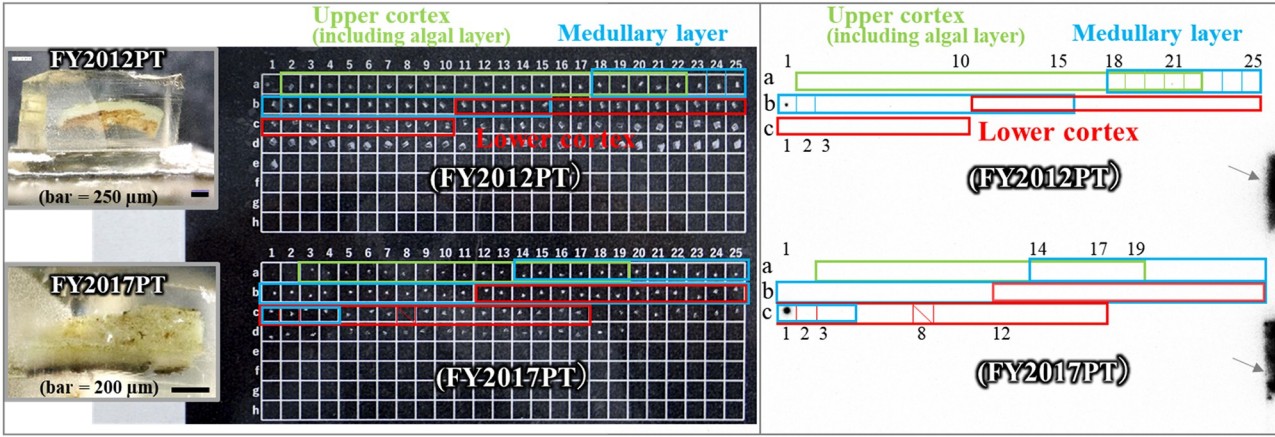

**Fig 7. Radiocaesium distributions in the tissue sections of FY2012PT and FY2017PT selected from fragments showing spot distributions in Fig 2.** The thallus tissues were prepared from the spot distributions "a" shown in Figs 1 and 2 for the FY2012PT and FY2017PT samples. Left panel: cross-sectional images of the FY2012PT (upper) and FY2017PT (lower) samples, as in Figs 3 and 5. Images of their tissue sections on the cell plate are also shown. Each tissue section was sliced from the surface side of the thallus tissue towards the lower cortex layer and laid on the cell plate in that order, the same as in Figs 3 and 5. Coloured frames are sections containing tissue. Green, aqua and red boxes indicate areas corresponding to the upper cortex (including the algal layer), medullary layer and lower cortex, respectively, which is also the case for the right panel. Overlapping boxes show that two layers were contained in an area. The sections were exposed for 24 hours on the IP. A shaded area, i.e., "c-8" in the FY2017PT plate, is missing a number. Right panel: autoradiograph images of tissue sections for the FY2012PT and FY2017PT samples. Spot distribution images were found in the "a-21" and "b-1" layers of the FY2012PT sample and in the "c-1" layer of the FY2017PT sample. Grey arrows indicate marks used to determine the location of the cell plate boxes.

## Discussion

### Radiocaesium distributions in lichen tissues

In this study, radiocaesium distribution patterns in lichen samples exhibited "broad" and "spot" distributions by autoradiography. In broad distributions, radiocaesium was localised in the brown pigmented parts (melanin-like substances) of the lower cortex in both the FY2012PT and the FY2017PT samples. We hypothesised that melanin-like substances were associated long-term retention of radiocaesium. Based on this hypothesis, quantum chemical calculations were performed and showed that the functional group of melanin-like substances can form complexes with $Cs^+$ ions.

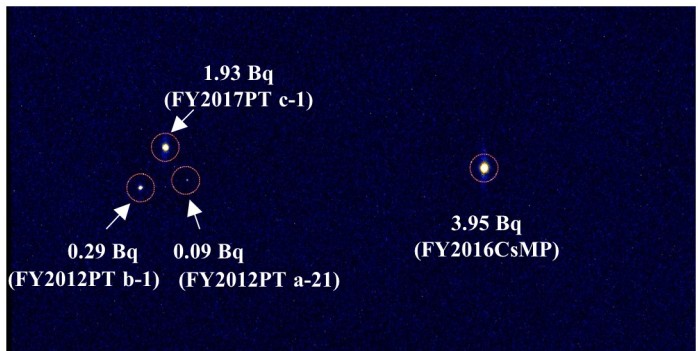

**Fig 8. Radioactivities in Cs-containing particles as estimated from autoradiography.** Tissue sections with Cs-containing particles (left side) were attached to the IP. The CsMP (FY2016CsMP) isolated from the parmelioid lichen by Dohi et al. [21] was used as a reference because the radioactivity was already determined (right side). The exposure time was 16 hours. The radioactivities in these Cs-containing particles were estimated.

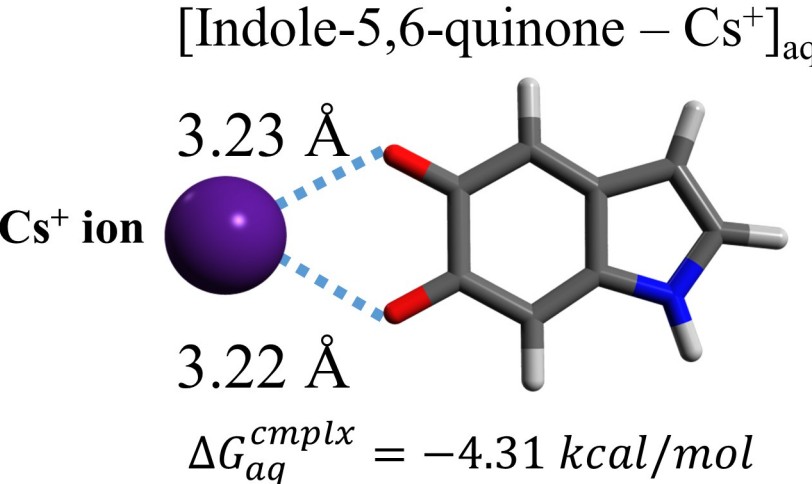

**Fig 9. Most stable structure for the Cs⁺ ion complex with indole from a melanin molecule in the aqueous phase.**
This typical functional compound of a melanin monomer, the indole group, was used for quantum chemical calculations. The distances from the Cs⁺ ion to the nearest oxygen atoms of the molecules are indicated. Each symbol shows element bonds: grey, C; white, H; blue, N; red, O.

Similar broad distributions of radiocaesium were also found in the leaves of bamboo plants and samples from chestnut trees collected in Fukushima prefecture within three years after the Fukushima nuclear accident, although some spotty distributions were also observed [37, 38]. Soudek *et al.* [39] studied $^{137}$Cs distributions in sunflower with hydroponic culture experiments using $^{137}$CsCl solution. Since $^{137}$Cs$^+$ ions are highly mobile and spread to give broad distributions within plant tissues, the autoradiogram showed an image where $^{137}$Cs was spread evenly throughout the leaves [39]. The broad distributions of radiocaesium in the above plant leaves collected in Fukushima resembled those in the experimental sunflower leaves, suggesting that radiocaesium would remain in an ionic form because of ion absorption. The broad distribution in the lichen samples also suggested that $^{137}$Cs may remain in a similar ionic form in the thallus.

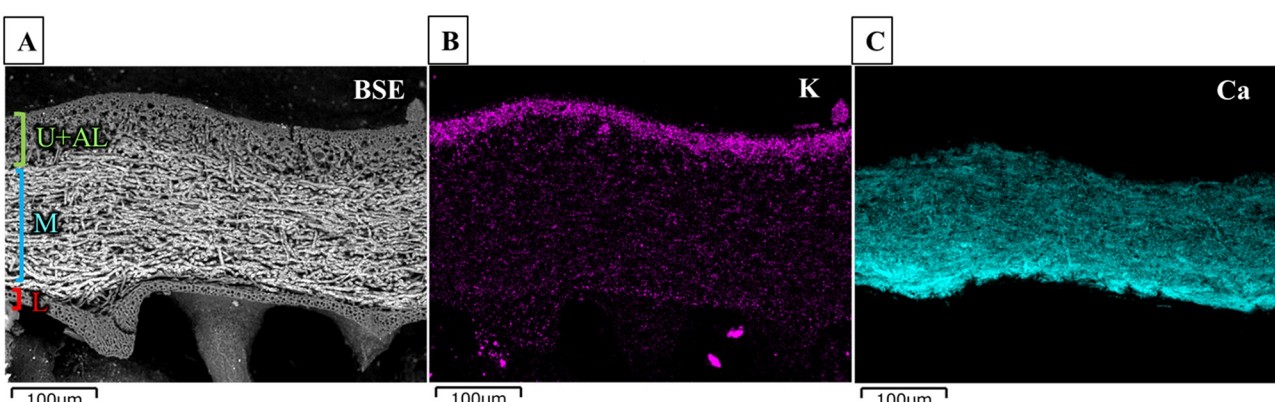

**Fig 10. Microscopic image and elemental analysis of the FY2012PT thallus cross-section.** The sample was analysed without any coating by SE-SEM with EDS. (A) Cross-sectional backscattered electron (BSE) image. U+AL, upper cortex including the algal layer; M, medullary layer; L, lower cortex. The M-layer appears white in the BSE image because it contains elements with higher atomic numbers than the surrounding elements. (B) Potassium and (C) calcium distributions were observed by EDS.

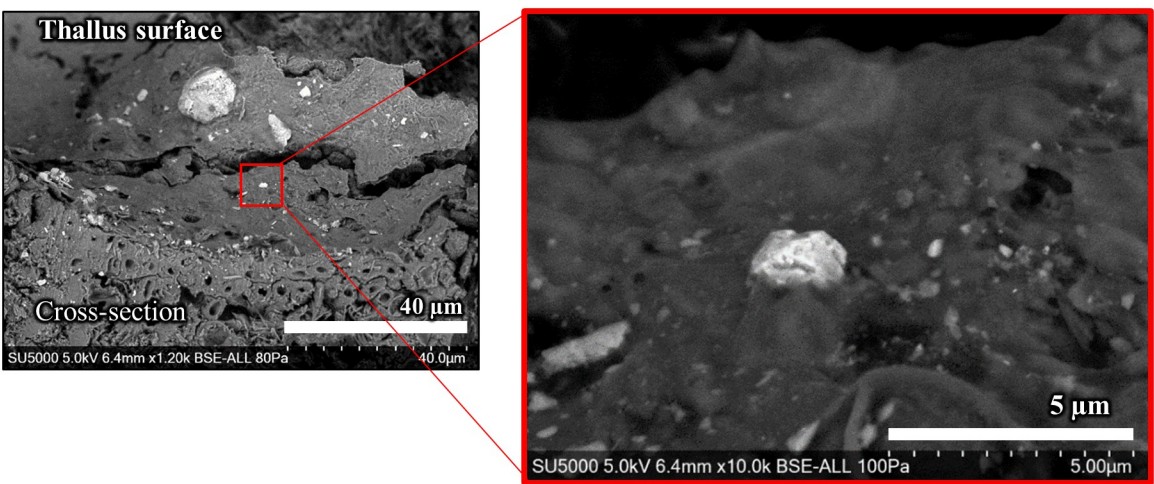

**Fig 11. BSE images of the lichen thallus of FY2012PT.** Left panel: micron-sized particles entrapped on the lichen thallus surface. Right panel: interaction between a micron-sized particle and the thallus surface. The sample was observed by SE-SEM without any coating. The right panel is a magnified image of the red boxed area in the left panel.

The *in situ* autoradiograms of the tissue sections revealed localisation of $^{137}$Cs in the lichen thallus and brown parts in the lower cortex. Typical parmelioid lichens have a well-developed lower cortex and are often strongly pigmented and melanised [25, 40]. Our sample species, the thallus of *P. tinctorum*, also had melanin-like pigmentation in the lower cortex [29], so the

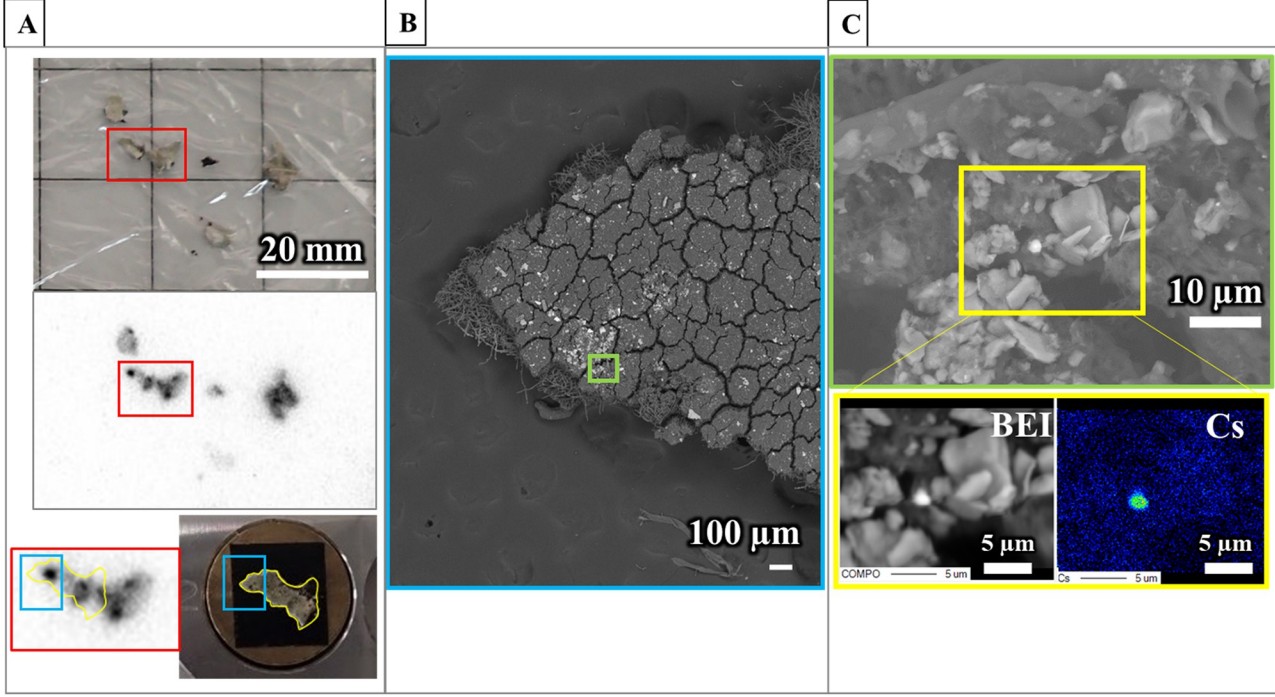

**Fig 12. Electron microscopic analysis for detecting a caesium microparticle on the lichen thallus surface.** The magnified areas in panels (A)–(C) are shown with corresponding coloured rectangles. (A) Autoradiograph image of FY2012PT. The fragments were exposed for 3 hours on the IP. The lower right panel shows the thallus sample on a brass sample table (10 mm φ × 10 mm height) for electron microscopic analysis. (B) BSE image of lichen thallus surface by FE-EPMA. A blue area in (A), particularly around a visible spot, was observed. (C) A micron-sized particle containing Cs was detected with FE-EPMA-WDS mapping analysis. The upper panel shows the green area in (B). A two-dimensional map analysis in the yellow area showed that the bright particle in the BSE image contained Cs. Other elements were also detected (See S13 Fig).

brown pigmented tissues in both the FY2012PT and the FY2017PT samples must be due to melanin substances. The locations of radiocaesium coincided with those of melanin substances in the thallus tissues, suggesting that melanin may be associated with retention of radiocaesium for two to six years.

Relationships between heavy metals and melanin pigments have been noted in lichen studies. McLean *et al.* [2] suggested that the apothecia of *Trapelia involuta* (Taylor) Hertel, a crustose lichen in Trapeliaceae growing on the U-containing mineral metazeunerite showed high melanin accumulation capacity by determining the U distribution on melanised apothecia with X-ray mapping. Then, they used FT-IR spectroscopy and showed that the pigments extracted directly from the apothecia were melanin or melanin-like compounds with humic acid structures [2]. Kasama *et al.* [3] used TEM analysis and reported that non-particulate U, Cu and Fe were distributed in the melanised exciple and epithecium of *T. involuta*. Fortuna *et al.* [25] compared the concentrations of Fe in six species of Parmeliaceae by ICP–AES and found that their Fe contents depended on the degree of melanisation and decreased in the order: heavily melanised on both sides of the cortex > heavily melanised on one side of the cortex > lightly melanised on one side of the cortex. Fogarty *et al.* [41] suggested that fungal melanin binds strongly to heavy metals (e.g., Cu, Cd, Ni, Pb, etc.) by summarising the following references: comparison of heavy metal contents (purified melanin from fungi > pigmented fungi > albino fungi) and the order of metal desorption from biomass (albino > pigmented > extracellular melanin). Since the melanin of lichens and fungi can bind various heavy metals, it may form complexes with Cs.

We determined the free energy for complex formation between the functional group (i.e., indole group) of melanin and a $Cs^+$ ion to assess the stability of the complex. The values obtained in this study were ca. $-4$ kcal $mol^{-1}$ for $\Delta G_{aq}^{cmplx}$ and 3.22 or 3.23 Å for bond lengths. Here, we compared the binding stabilities of $Cs^+$ ions with lichen substances: secondary metabolites and hydroxyindole based on melanin monomers. The computational calculations in Suno *et al.* [28] showed that $Cs^+$ ions could form complexes with the secondary metabolites, atranorin, lecanoric acid, usnic acid and protocetraric acid, all of which are produced in parmelioid lichens (e.g., *P. tinctorum*, *F. caperata*, etc.). The previously reported energies for complexation with secondary metabolites ranged from $-2.01$ to $-6.18$ kcal $mol^{-1}$ for $\Delta G_{aq}^{cmplx}$ and the bond lengths ranged from 3.12 to 3.44 Å [28]. The similarities of the formation energies for $Cs^+$ ion complexes in the aqueous phase and their bond distances to those for the secondary metabolites suggested that $Cs^+$ ions bind with melanin monomers and form complexes with similar degrees of stabilities.

Comparative studies carried out by Bruenger *et al.* [42] suggested that melanin synthesised *in vitro*, which originated from L-DOPA, formed complexes of alkali metal ions by ion exchange because $Cs^+$ uptake was limited by the presence of $Na^+$, $K^+$ and $Mg^{2+}$. However, the stability of $Na^+$ in natural eumelanin pigment was confirmed by a discharge experiment; density functional theory calculations combined with spectroscopic evidence showed that melanin has a porphyrin-like layered structure that affects $Na^+$ stabilisation by sandwiching it between layers located above and below the ion [43]. Hence, alkali-metal ions may insert into melanin polymers, such as porphyrin-like structures, which leads to retention of the stabilised ions.

According to a simulation study reported by Katata *et al.* [44], highly radioactive contamination occurred around the FDNPS from 15 to 16 March 2011 and was caused by rainfall, plume movements, etc. Rainwater with dissolved radiocaesium fell to the ground as wet deposition and contributed to high dose rates in that contaminated area [44]. Caesium-137 ions seem to have migrated through the thallus surface (upper layer), medullary layer and lower cortex in the thallus of *P. tinctorum*, or permeated those tissues at the same time.

However, radiocaesium distributions were not observed by autoradiography in the upper cortex and medullary layer tissue sections in either FY2012PT or FY2017PT. These Cs accumulation locations in lichen tissue layers provided additional clues for the mechanism of $^{137}$Cs uptake in lichens. Suno *et al.* [28] showed that Cs$^+$ ions can form complexes with metabolites such as oxalic acid, atranorin (within the upper layer) and lecanoric acid (within the medullary layer), although no specific Cs selectivity was observed from among a number of alkali-metal cations (Li$^+$ >Na$^+$ >K$^+$ >Rb$^+$ >Cs$^+$). Our electron microscopic analyses of tissue cross-section showed K accumulation in the upper cortex layer and Ca in the medullary layer of the FY2012PT thallus. Since the ion-exchange mechanism is known to involve metal accumulation [7], ion exchange of Cs$^+$ for K$^+$ may occur in the upper cortex after Cs deposition from the atmosphere and/or rain. Other studies have found that, gyrophoric acid, which is the main metabolite of *T. involuta*, does not seem to be responsible for binding of the heavy metals U, Cu and Fe, since their positional distributions did not coincide with the main thallus body producing the secondary metabolite [2, 3]. Calcium oxalate crystals in *T. involuta* were also investigated with FE-SEM EDS by Kasama *et al.* [3], but no heavy metals were detected. This was also consistent with our results. Calcium, which seems to form complexes with metabolite substances on the hyphal surface, was detected in the medullary layer of the FY2012PT thallus, but radiocaesium was not detected within the layer. Therefore, metabolites in the upper cortex and the medullary layers of *P. tinctorum* did not bind a significant number of Cs$^+$ ion. This is the first showing that radiocaesium localised in the lower cortex, and assessing the stability of Cs$^+$ ion complex formation with lichen melanin-like substances by combining our novel approach, *in situ* microscale observations by autoradiography and quantum chemical calculations.

Thus, we suggest that steric complexation of Cs$^+$ ions may be due to peculiarities in the molecular structure of melanin (e.g., porphyrin-like multimer structures) and result in stable retention of radiocaesium. Rassabina *et al.* [45] succeeded in extracting melanins directly from *Cetraria islandica* and *Pseudevernia furfuracea*, and identified them as allomelanin. In fungi, two kinds of melanin, i.e., eumelanin and allomelanin, are known to react to hostile conditions and protect intracellular homeostasis [41]. Since our calculations were focused on a monomeric model for eumelanin, a future study is needed to examine a monomeric counterpart of allomelanin. However, we can expect similar results to form this study because the structure also includes multiple hydroxy groups and oxygen atoms, as does the monomeric structure of eumelanin. Information on the chemical structures and characteristics of lichen melanin is still limited, and further study is needed to explore the polymeric and three-dimensional structures of melanin.

## Distribution of radiocaesium in particulate form

Regarding spot distributions in lichen samples, combined autoradiography and electron microscopy revealed that Cs-containing particle was present on the thallus surface of the upper cortex layer in FY2012PT. The $^{134}$Cs/$^{137}$Cs activity ratio was 1.01±0.11, and the elemental compositions were consistent with those of CsMPs derived directly from the FDNPS. Grassy particles with activity ratios ranging from 0.90 to 1.20, and diameters ranging from 1 to 10 μm commonly contained Si, O, Fe, Zn, Sn, etc and were recognised as CsMP type-A particles [46–48]. These specific particles were characteristic of the Fukushima accident and were clearly different from Cs adsorbed by soil mineral particles [49], showing that the Cs-containing particle on the thallus surface was CsMP.

Other Cs-containing particles were also found to have spot distributions in tissue sections of both the FY2012PT and the FY2017PT samples, and three spot distributions were located

around the medullary layer. These three spot distributions and other spot distributions, including that of isolated CsMP in the FY2012PT thallus (mentioned above) and that of FY2016CsMP sample, showed similar spectral shapes, indicating Cs-containing particles. CsMPs have been shown to have radioactivity levels above 0.4 Bq [32, 46]. In contrast, no other particles sorbing radiocaesium (e.g., soil minerals, organic matter or aggregated fine minerals) had radioactivities that exceeded 0.2 Bq [32, 50]. Notably, Cs-containing particles with radioactivity levels below 0.2 Bq were indistinguishable from CsMPs or other Cs-containing particles [32]. Thus, the Cs-containing particles within the b-1 layer in the FY2012PT sample (0.29 Bq) and the c-1 layer in the FY2017PT sample (1.93 Bq) indicated to be CsMPs. The particle within the a-21 layer of the FY2012PT sample (0.09 Bq) was not distinguished as a CsMP.

In terms of CsMP localisation, CsMPs extracted and isolated directly from parmelioid lichens showed strong autoradiographic spots similar to those seen here, although no information was reported on CsMP localisation in the thallus because the tissues were degraded [21]. Submicron to >100 μm airborne particles containing metals were detected physically and/or appeared adhered to the thallus surfaces of lichens and were also detected in the medulla [10, 24]. Although it is generally difficult to obtain time records indicating "when the particles were emitted" unless they have source-specific elements, CsMPs originating from the Fukushima accident have distinctive compositions and structures. In other words, CsMPs are tracers providing a time record of 2011. CsMPs were observed on/in the thallus even two and six years after the Fukushima accident, suggesting that the micron-sized particles remained stable on/in the thallus.

Our findings suggested a mechanism for how CsMPs could remain in the thallus after such a long period. In electron microscopy images, micron-sized particles appeared to adhere to the thallus surface tissue of sample FY2012PT. This adhesion may have arisen because the upper cortex layer of parmelioid lichens comprises a gelatinized and conglutinated structure of mycelium [51], which arises from the extracellular matrix (ECM) of polysaccharides in the hyphal cell wall, including secondary metabolic products, that link the hyphae together and form an adhesive layer [52]. Additionally, K was predominantly found in the upper area of the upper cortex layer in FY2012PT, indicating that hyphal growth was more active on the thallus surface area [53]. Particles trapped on the thallus surface that were not physically shed may be embedded in this ECM, which would contribute to the long retention period.

Particles that physically reached the tissues corresponding to the medullary layer and lower cortex, e.g., by surface cracks, pores and/or pseudocyphellae, have been shown to remain in the interwoven spaces that occupied approximately 18% of the thallus volume of hyphae [10, 22, 54]. This phenomenon was supported by detection of radiocaesium particles on/in the thallus in the same sample FY2012PT. Therefore, we suggest that micron-sized particles that remained for a long time without being physically washed out were trapped by development of the adhesive layer.

## Conclusion

In this study, we used combined *in situ* microscale observations by autoradiography and quantum calculations for the first time to determine radiocaesium distributions in thallus tissues of *P. tinctorum* collected two and six years after the Fukushima nuclear accident. We also directly identified radiocaesium particles on and in thallus tissues and determined their locations with autoradiography and electron microscopic analyses. We suggest that melanin substances in the lower cortex may lead to stable complexation of $Cs^+$ ions and contribute to long-term retention. With regard to radiocaesium particles deposited on thallus surface tissues,

development of the adhesive layer in the upper cortex, and not just physical retention on/in the upper cortex and in the medullary layer, is involved in retention of Cs particles.

Our findings may be helpful for estimating radiocaesium accumulation capacities and retention times when lichen species are applied as radiocaesium biomonitors in the future. In addition, CsMPs which remain as a "recording medium for Cs" in lichens, are expected to determine the amount of CsMPs deposited in the area, even after a lapse of several years from the Fukushima accident.

## Supporting information

**S1 Fig. Thallus tissue sections of the FY2017PT.** The tissue sections were prepared from the broad distribution "a" of the FY2017PT sample in Figs 1 and 2. The lower right panel image shows cross-sectional image same as Fig 5A. Autoradiograph images of these sections were shown in Figs 5 and 6, and S8–S12 Figs. Coloured frames are sections containing tissue. Green, aqua and red boxes indicate the areas corresponding to the upper cortex including the algal layer (U+AL in cross-sectional image), medullary layer (M) and lower cortex (L), respectively. Overlapping boxes indicate that two layers were contained in the frames. The "a-4" and "b-2" are missing numbers. The "b-3" layer contains 2 sections. Bar = 200 μm.
(TIF)

**S2 Fig. Signal intensity patterns of the spot distributions in autoradiograms of the Cs-containing particles.** The red line "a" shows the pattern of the CsMP trapped in lichen (FY2016CsMP), which was previously determined by Dohi *et al*. [21]. The CsMP sample was exposed for the 16 hours on the IP. The blue line "b" shows the pattern of the Cs-containing particle in Fig 12. The linear intensity through the centre (x-axis aligned with 0) of each autoradiogram spot is shown as an example. The x-axis indicates linear distance of the distribution (μm), and the y-axis indicates the PSL signal intensity values (counts) in both graphs.
(TIF)

**S3 Fig. Radiocaesium distributions of "c-3" layer in the tissue sections of the FY2012PT sample showing broad distribution by autoradiography.** The radiocaesium distributions of the "c-3" to "c-8" layers in the FY2012PT sample. The "c-4" layer was excluded here (see Fig 4). S3–S7 Figs show c-3 to c-8 layers, respectively. (A) Overlapping image of the autoradiography result and tissue section image. Background counts were deducted from all count values. Each colour indicates the range of the PSL signal intensities, as shown in the legend, which also applies to panel (B). (B) Three-dimensional image of (A) in a 1 mm square area. (C) Image of tissue section taken with the digital microscope. The brown parts are pigmented tissue (yellow arrow), white parts are hyphae (white arrow), and transparent parts are resin (aqua arrow). Hereinafter the same within S3–S7 Figs. Cutting marks from the glass knife appeared as a white linear pattern on the resin.
(TIF)

**S4 Fig. Radiocaesium distributions of "c-5" layer in the tissue sections of the FY2012PT sample.**
(TIF)

**S5 Fig. Radiocaesium distributions of "c-6" layer in the tissue sections of the FY2012PT sample.**
(TIF)

**S6 Fig. Radiocaesium distributions of "c-7" layer in the tissue sections of the FY2012PT sample.**
(TIF)

**S7 Fig. Radiocaesium distributions of "c-8" layer in the tissue sections of the FY2012PT sample.**
(TIF)

**S8 Fig. Radiocaesium distributions of "b-4" layer in the tissue sections of the FY2017PT sample showing broad distribution by autoradiography.** The radiocaesium distributions of the "b-4" to "b-9" layers in the FY2017PT sample. The "b-5" layer was excluded here (see Fig 6). S8–S12 Figs show b-4 to b-9 layers, respectively. (A) Overlapping image of the autoradiography result and tissue section image. The background counts were deducted from all count values. Each colour indicates range of the PSL signal intensities, as shown in the legend, which also applies to panel (B). (B) Three-dimensional image of (A) in a 1 mm square area. (C) Image of tissue section taken with the digital microscope. The brown parts are pigmented tissue (yellow arrow), and transparent parts are resin (aqua arrow). Hereinafter the same within S8–S12 Figs. Cutting marks from the glass knife appeared as a white linear pattern on the resin.
(TIF)

**S9 Fig. Radiocaesium distributions of "b-6" layer in the tissue sections of the FY2017PT sample.**
(TIF)

**S10 Fig. Radiocaesium distributions of "b-7" layer in the tissue sections of the FY2017PT sample.**
(TIF)

**S11 Fig. Radiocaesium distributions of "b-8" layer in the tissue sections of the FY2017PT sample.**
(TIF)

**S12 Fig. Radiocaesium distributions of "b-9" layer in the tissue sections of the FY2017PT sample.**
(TIF)

**S13 Fig. Elemental mapping of a Cs-containing particle.** This analysis was carried out to determine a Cs-containing particle (red circle) and its co-existing elements on the thallus surface of the FY2012PT sample using FE-EPMA-WDS.
(TIF)

## Acknowledgments

We would like to express our gratitude to Dr. Masamichi Shiono of Hitachi High-Tech Corporation, and Drs. Norihisa Mori and Tomohiro Mihira of JEOL Ltd. for their technical assistance and advice in electron microscopic analyses; to Mr. Hisaya Tagomori, Dr. Yasuo Ishii and the Fukushima Radiation Measurement Group of JAEA for their contributions to the measurement of radioactivity; and to Drs. Taiga Okumura and Toshihiro Kogure at the University of Tokyo for helpful advice on digital micrograph analysis. We are grateful to Drs. Yoko Kokubu, Kenichi Yasue, Tsuneari Ishimaru, Hiroki Amamiya and Shinji Makita for their kind assistance during field investigations in JAEA (Toki city).

## Author Contributions

**Conceptualization:** Terumi Dohi.

**Formal analysis:** Terumi Dohi, Kazuki Iijima, Masahiko Machida, Hiroya Suno, Shigeru Kimura, Futoshi Kanno.

**Funding acquisition:** Terumi Dohi.

**Investigation:** Terumi Dohi, Yoshihito Ohmura, Shigeru Kimura, Futoshi Kanno.

**Methodology:** Terumi Dohi, Kazuki Iijima, Masahiko Machida, Hiroya Suno.

**Project administration:** Terumi Dohi.

**Supervision:** Kazuki Iijima.

**Visualization:** Terumi Dohi, Kazuki Iijima, Masahiko Machida, Hiroya Suno, Shigeru Kimura, Futoshi Kanno.

**Writing – original draft:** Terumi Dohi, Kazuki Iijima, Masahiko Machida.

**Writing – review & editing:** Terumi Dohi, Kazuki Iijima, Masahiko Machida, Hiroya Suno, Yoshihito Ohmura, Kenso Fujiwara.

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
