## [Decision Letter · Decision Letter 0]

19 May 2022

PONE-D-22-09673Accumulation mechanisms of radiocaesium within lichen thallus tissues by means of in situ microscale localisation observationPLOS ONE

Dear Dr. Dohi,

Thank you for submitting your manuscript to PLOS ONE. After careful consideration, we feel that it has merit but does not fully meet PLOS ONE’s publication criteria as it currently stands. Therefore, we invite you to submit a revised version of the manuscript that addresses the points raised during the review process.

We look forward to receiving your revised manuscript.

Kind regards,

Nathalie A. Wall, Dr.

Academic Editor

PLOS ONE

Journal Requirements:

"This research was supported by Grant-in-Aid for Challenging Exploratory Research of JSPS (no.16K12627) operated by TD. The funder had no role in study design, data collection and analysis, decision to publish, or preparation of the manuscript. They provided only financial support in the research materials in this study. Nuclear Engineering Co., Ltd. employed SK, and Pesco Co., Ltd. employed FK, these commercial affiliations did not play a role in this study."

Reviewers' comments:

Reviewer's Responses to Questions

**Comments to the Author**

1. Is the manuscript technically sound, and do the data support the conclusions?

Reviewer #1: Yes

Reviewer #2: Yes

2. Has the statistical analysis been performed appropriately and rigorously? 

Reviewer #1: Yes

Reviewer #2: Yes

3. Have the authors made all data underlying the findings in their manuscript fully available?

Reviewer #1: Yes

Reviewer #2: Yes

4. Is the manuscript presented in an intelligible fashion and written in standard English?

Reviewer #1: No

Reviewer #2: Yes

5. Review Comments to the Author

Reviewer #1: Reviewer comments for the manuscript “Accumulation mechanisms of radiocaesium within lichen thallus tissues by means of in situ microscale localisation observation”

The topic of this manuscript is interesting and it gives new insight on localisation of Cs in lichen after accumulation from the atmosphere, and also it combines this phenomenon to one possible fate of 137Cs-rich microparticles, which were released in the Fukushima NPP accident. The presented study itself appears to be of good quality and have a solid basis. Manuscript contains many figures, which are informative. As a published article, this manuscript would benefit researchers working in the fields of environmental radioactivity, radioecology, and environmental contamination, at least.

I’m not qualified to check the English of this manuscript, but I can see that it needs a thorough language revision, in order to make the text reader-friendly and for easier understanding. I haven’t listed all words and sentences which need editing, because there are so many of them, and I can’t give instructions for correcting all of them. I’m convinced that after a language check by a native English speaker these grammatical errors will disappear and this check would improve greatly the clarity of the manuscript.

My detailed reviewer comments are below and they all are technical ones. After full language check, I can then consider recommending this manuscript to be published in Plos One.

Abstract

line 19: Many lichen are -> Many lichens are

line 26: The radiocaesium localised -> The radiocaesium was localised

Introduction

line 52: please use “a” as a unit for year, so no “y” should be used for the half-life

line 75: “those” -> I think that this would work better, if replaced with “which”

Materials and Methods – Samples

line 98 (and possibly elsewhere, please check): “y” should be replaced with “a” as an official unit for a year

line 104: “where” -> “which” would make this sentence more fluent

Autoradiography of lichen samples and selection of thallus tissue

lines 125-127: I don’t understand this sentence in its current form. It is probably just some word, which should be replaced or added, and then this sentence would open, but now it is difficult to read. I’m sure that if and when the text is revised by a native English speaker and he/she can negotiate with the authors, what they are trying to express here, then this sentence will be fixed. I’m looking forward to seeing the next version.

line 28: thallus tissue -> thallus tissues

Preparation of thallus tissue sections

lines 149 and 157: 4°C -> 4 °C

line 153: please remove “was prepared” (duplicate is on the previous row)

line 159: 60°C -> 60 °C

Evaluation of the radiocaesium distribution in thallus tissue sections

line 177: were estimated -> was estimated

Quantum chemical calculations

lines 181-182: “Cs+ ion and biological substance contained tissue which radiocaesium distribution was observed” this would be clearer as reorganised: “Cs+ ion and a biological substance containing tissue, where radiocaesium distribution was observed”

lines 189-195: It would be good to split this very long sentence to four sentences, as follows: “Our quantum computational technique used a stepwise approach. The first step combined the semiempirical PM6 (parametric model 6) method and the multicomponent artificial force-induced reaction (MC-AFIR) method to screen for stable reaction-product structures from among several candidates. The second step subsequently used density functional theory (DFT) to optimise the screened candidate structures at the ab initio quantum mechanical level. The third and final step combined DFT with the polarisable continuum model (PCM) to take into account solvation effects.”

Gamma-ray measurements for radiocaesium

line 227: “The 134Cs and 137Cs radioactivity” -> either “The radioactivities of 134Cs and 137Cs” or “The 134Cs and 137Cs radioactivities”

line 236: sec -> s

the same line: single and multiple are mixed here. It should be “The particles were measured…” due to multiple form in the later part of the sentence.

Results

lines 270-271 (and later at lines 295-296, 316-317 similar cases): Figure caption of figure 3. What is the meaning of this sentence? Are the authors trying to say, e.g., that the broad distribution is seen both in figures 1 and 3? In that case, the sentence should be for example: “The radiocaesium distributions in the tissue sections of the FY2012PT thallus showing broad distribution, which was determined also by autoradiography in Fig 1.” Or “The radiocaesium distributions in the tissue sections of the FY2012PT thallus showing broad distribution, which can be also seen in autoradiograph in Fig 1.” Or similar. The problem is again the use of English and it will be fixed if a Native speaker will revise this manuscript.

line 298: please remove microtome, as it is a duplicate in the same sentence

lines 343-344: “This result is evidenced Cs+ ion can form complex with functional group of melanin-like substances in neutral aquatic.” This sentence lacks some words and it should be written in a clearer way, e.g., “This result proves that Cs+ ion can form a complex with a functional group of melanin-like substances in a neutral aquatic phase.”

lines 370-371: “The thallus surface was also examined by SE-SEM to determine how present micron-sized particles.” This is now difficult to understand, I suggest a change: “The thallus surface was also examined by SE-SEM to determine the presence of micron-sized particles.”

lines 383-384: “…are helpful in comparison the characteristics of CsMPs reported previous studies.” This should be modified for improving clarity, e.g., “…are helpful in comparing the characteristics of CsMPs reported in the previous studies.”

line 404: What does this sentence mean? Radiocesium = can be either 134Cs or 137Cs, so which isotope was in question, or was it both?

Discussion

line 409: radiocaesium localised -> radiocaesium was localised

line 419: “autoradiogram showed an image spread evenly throughout the leaves” -> I guess that it was not an image which was spread evenly throughout the leaves, rather it was Cs? Therefore, I suggest a change for a better clarity: “autoradiogram showed an image where caesium was spread evenly throughout the leaves”

line 456: “The similarity of the energies of formation for Cs+ ion complexes” -> “The similarity of the formation energies for Cs+ ion complexes”

line 464: effects -> affects

line 467: “was” is unnecessary here

line 474: These Cs accumulation location -> These Cs accumulation locations

line 490: first -> the first

the same line: whose stability is assessed here? This sentence is again a bit unclear. If the purpose is to assess the stability of Cs isotopes or ions, it should be written here.

line 508 (and previously in Gamma measurement sections): is it really and isotopic ratio of 134Cs/137Cs (synonyms are atom ratio and mass ratio) or activity ratio 134Cs/137Cs (is obtained simply from the radioactivities of the both isotopes)? Please check this as it is important to use a correct ratio name.

line 520: [32, 50,] -> [32, 50]

line 532: are tracer -> are a tracer/are tracers

Conclusions

line 562: “which” can be removed as unnecessary

line 563: remains -> remain

the same line: “are expected to be examined the amount of CsMPs…” this lacks a preposition and could be improved, either “are expected to be examined for the amount of CsMPs…” or “are expected to be examined due to the amount of CsMPs…”

Reviewer #2: The paper deals with radioceasium including CsMPs dispersed by the Fukushima accident by using combined in situ autoradiography and quantum calculations. The new method reveals radiocaesium distributions in lichen thallus tissues, and their accumulation mechanisms.

This is an interesting observation. And the manuscript is well written, so the reviewer would recommend it for acceptance after a few points.

Minor comments.

L106　134Cs and 137Cs were not detected in the background sample.

Sentences in the same context is written in the Result section (L401), so the reviewer thinks the sentence can be deleted. If the authors want to remain the sentence, elemental symbol at the beginning of a sentence should be spell out. i.e. 134Cs -> Cesium-134

L143-175 In the autoradiography study, background signals from areas without samples were subtracted from the values from each section area. The subtracted value was taken as the radiation from the specimen.

These sentences seem to be not clear. Background signals should be subtracted as counts (values) per time; not accumulated counts (values). The sentences should be clear more.

L234-236 The radiocaesium (134, 137Cs) activities were determined based on standard point sources, CZ402 for 134Cs and CS402 for 137Cs (Japan radioisotope association, Tokyo, Japan) [21].

The standard point sources (CZ402 and CS402) have 6mm diameter. On the other hands, CsMPs have a few µm diameter. Do the standard point sources have enough accuracy to determine the radioactivity of CsMPs?

6. PLOS authors have the option to publish the peer review history of their article (what does this mean?). If published, this will include your full peer review and any attached files.

Reviewer #1: No

Reviewer #2: No

---

## [Author Response · Author response to Decision Letter 0]

16 Jun 2022

Dear Dr. Nathalie A. Wall and Reviewer persons,

We would like to thank all academic editor and reviewer persons for taking time out of your precious time to peer review and review comments.

We revised our manuscript based on your kind comments and advice.

In addition, we checked English in the revised manuscript throughout, again.

We hope that we have answerd your comments in this manuscript and in "Response to reviewers (2 files)".

Yours sincerely,

Terumi Dohi

---

## [Editor Report · Decision Letter 1]

23 Jun 2022

Accumulation mechanisms of radiocaesium within lichen thallus tissues determined by means of in situ microscale localisation observation

PONE-D-22-09673R1

Dear Dr. Dohi,

We’re pleased to inform you that your manuscript has been judged scientifically suitable for publication and will be formally accepted for publication once it meets all outstanding technical requirements.

Kind regards,

Nathalie A. Wall, Dr.

Academic Editor

PLOS ONE
---

## [Editor Report · Acceptance letter]

29 Jun 2022

PONE-D-22-09673R1 

Accumulation mechanisms of radiocaesium within lichen thallus tissues determined by means of *in situ* microscale localisation observation 

Dear Dr. Dohi:

I'm pleased to inform you that your manuscript has been deemed suitable for publication in PLOS ONE. Congratulations! Your manuscript is now with our production department. 

Kind regards, 

on behalf of

Prof. Nathalie A. Wall 

Academic Editor

PLOS ONE